# Caprine Arthritis Encephalitis Virus Disease Modelling Review

**DOI:** 10.3390/ani11051457

**Published:** 2021-05-19

**Authors:** Karina Brotto Rebuli, Mario Giacobini, Luigi Bertolotti

**Affiliations:** Dipartimento di Scienze Veterinarie, Università degli Studi di Torino, Largo Paolo Braccini 2, 10095 Torino, Italy; karina.brottorebuli@unito.it (K.B.R.); luigi.bertolotti@unito.it (L.B.)

**Keywords:** CAE, CAEV, SRLV, epidemiological modelling, statistical modelling, diary production modelling

## Abstract

**Simple Summary:**

Mathematical modelling is used in disease studies to assess their economical impacts, as well as to better understand the epidemiological dynamics of the biological and environmental factors associated with disease spreading. For an incurable disease such as Caprine Arthritis Encephalitis, this knowledge is extremely valuable. However, the application of modelling techniques to study this disease has not been significantly explored in the literature. The purpose of the present work was to review the published studies, highlighting their scope, strengths and limitations, as well to provide ideas for future modelling approaches for this disease. The reviewed studies were divided into two major themes. The first is epidemiological modelling, which use mathematical models which equations describe the disease dynamics over time. Inside this group, the articles differ in considering or not considering the sexual transmission component. The second major theme is statistical modelling, which correlates the disease with biological and environmental factors to quantify its risks and impacts. Inside this group, the articles include models for dairy production, for risk factors of the disease and for Caprine Arthritis Encephalitis being a risk factor for other diseases. Finally, the present work concludes with further suggestions for modelling studies on Caprine Arthritis Encephalitis.

**Abstract:**

Mathematical modelling is used in disease studies to assess the economical impacts of diseases, as well as to better understand the epidemiological dynamics of the biological and environmental factors that are associated with disease spreading. For an incurable disease such as Caprine Arthritis Encephalitis (CAE), this knowledge is extremely valuable. However, the application of modelling techniques to CAE disease studies has not been significantly explored in the literature. The purpose of the present work was to review the published studies, highlighting their scope, strengths and limitations, as well to provide ideas for future modelling approaches for studying CAE disease. The reviewed studies were divided into the following two major themes: Mathematical epidemiological modelling and statistical modelling. Regarding the epidemiological modelling studies, two groups of models have been addressed in the literature: With and without the sexual transmission component. Regarding the statistical modelling studies, the reviewed articles varied on modelling assumptions and goals. These studies modelled the dairy production, the CAE risk factors and the hypothesis of CAE being a risk factor for other diseases. Finally, the present work concludes with further suggestions for modelling studies on CAE.

## 1. Introduction

Caprine Arthritis Encephalitis (CAE) disease is a world-wide goat infectious disease [1,2,3,4,5,6,7] caused by Small Ruminant Lentiviruses (SRLV). Early phylogenetic studies suggested that SRLV can be divided into five genetic groups, A to E [6]. Genotypes C and D as well subtypes A5 to A7 circulate only in goats; the subtype A2 circulates only in sheep, while subtypes A1, A3, A4, A6, B1 and B2 have been found in both species [8]. In goats, SRLV have a long incubation time and symptoms may be evident in only 10% of goats from a SRLV-infected herd. The clinical symptoms of CAE are arthritis, indurative mastitis and, more rarely, encephalomyelitis in juvenile [9,10]. Occasionally, the infected ones may develop chronic interstitial pneumonia and progressive dyspnea [11]. The SRLV can be transmitted from infected animals with or without clinical signs [12], mainly by the colostrum ingestion (vertical transmission) and by direct contact (horizontal transmission) in pasture or by animals trading [13]. Other routes like the intrauterine transmission [14] and sexual contact [15] are considered neglected or minor.

Since there is no vaccine or cure for CAE, prophylactic measures are very important. They include: (i) separation of kids from seropositive dams immediately at birth; (ii) the use of heat-treated colostrum or SRLV-negative colostrum; (iii) increase of culling rate; and (iv) periodic serological surveillance and immediate segregation of seropositive goats [6]. The comprehension of the epidemiological dynamics and of the biological and environmental factors that are associated with the disease spreading is crucial to improve the application of these prophylactic measures. In epidemiological field, modelling techniques have been proved to be helpful, both by providing epidemic predictions and by giving indications of how much each transmission route and biological feature of the disease can contribute to the infection contention, prevention or eradication. Additionally, modelling has been used also to quantify and predict the economic impact of infection diseases.

Mathematical and statistical models are a set of rules that associates quantitatively the observed variables and the outcomes of a phenomenon. They can be used to make predictions of the outcome given the input variables or, in the opposite way, to make inferences about the phenomenon variables given its outcome. Importantly, any model has its own assumptions and the limits within which it is valid. That is why there is no the definitely best model for every problem [16], and many modelling strategies have been developed.

When using mathematical models, the phenomenon is described by the equations that quantify its processes and transformations on time. A very known example is the Susceptible-Infected-Recovered (SIR) model [17] in the epidemiological field. In this model, the individuals in the population are divided into compartments according to the their status of the disease. This division means that for each of these compartments an ordinary differential equation (ODE) is defined. These ODEs represent the biological laws that explain and quantify the infection spreading and they allow the model to represent the epidemics dynamic continuously on time. The very basic SIR model is defined by:(1)$$\left\{\begin{array}{c} \frac{dS}{dt}\;=\;-\beta SI\\ \frac{dI}{dt}\;=\;\beta SI-\gamma R\\ \frac{dR}{dt}\;=\;\gamma I \end{array}\right.$$
where *S* is the susceptible, *I* is the infected and *R* is the recovered compartments (number of individuals at an instant of time); β is the infection rate; and γ is the recovery rate. In mathematical modelling approach, once the equations of the model and the initial conditions are defined, the mathematical behaviour of the system is deterministic. It is studied by techniques like the analysis of its equilibra points (set of solutions *M* such that x(0) in *M*⇒x(t)∈M∀t∈R), numerical simulations, amongst others. As examples of the application of mathematical epidemiological models (referred just as epidemiological models from now and on, in the sake of the simplicity) in veterinary epidemiology, Gaucel et al. [18] and Courcoul & Ezzano [19] used epidemiological models for studying the spread of bovine viral diarrhoea virus within and between herds, Babayani et al. [20] used an epidemiological model to predict the haemonchosis disease in sheep on a commercial farm in South Africa, and Murai et al. [21] studied through epidemiological modelling the conditions to control porcine epidemic diarrhea, a disease that can be damaging for pig producers. Apart from the applications in veterinary field, there is a profusion of epidemiological models for human diseases too. Páez Chávez et al. [22] proposed an epidemiological model for dengue transmission in order to study the effectiveness of the control strategies. Recently, Srivastava et al. [23] explored a set approximated solutions for the fractional epidemiological model of the Ebola virus epidemics. Enrique Amaro et al. [24] and Gevertz et al. [25] applied variations of epidemiological models for analysis of the SARS-Cov-2 pandemic spreading.

Statistical models are very different. They use functions that represent probability spaces to build the relationship between the input (independent) and output (dependent) variables. While mathematical modelling analysis consists in studying the behaviour of the system defined by the ODEs, the analysis of statistical models consists in finding the values of the model parameters. The statistical theory used to support the specification of the model and the strategy to assess its parameters will define if the analysis will be frequentist or Bayesian. The simplest case of a statistical model is a linear regression, a linear relationship between the outcome and one or more input variables plus a random error. For example, Selvaraju et al. [26] used a multiple linear regression model for forecasting sheep bluetongue disease outbreaks. As in any statistical modelling analysis, the general stochastic model (Equation (Equation 2)) was fitted to the data in order to define the model coefficients to build the specific stochastic model for the problem (Equation (Equation 3)).
(2)$$y\;=\;\beta_{0}+\beta_{1}W_{1}+\beta_{2}W_{2}+\beta_{3}W_{3}+\beta_{4}W_{4}+\beta_{5}W_{5}+\beta_{6}W_{6}+\epsilon$$
where *y* is the dependent variable; $\beta_{0}$ is the intercept (constant); $\beta_{0}$ to $\beta_{6}$ are the coefficients of the $W_{1}$ to $W_{6}$ variables: monthly mean maximum temperature (°C), monthly mean minimum temperature (°C), relative (%) humidity at 8h30 IST, relative (%) humidity at 17h00 IST, monthly total rainfall (mm) and monthly mean wind speed (km/h); and ϵ is the residual random error. The final form of the model resulted as: (3)$$y\;=\;220.453+\left(-8.922\right)W_{1}+5.358W_{2}+0.183W_{3}+\left(-0.846\right)W_{4}+0.049W_{5}+\left(-2.412\right)W_{6}+\epsilon$$

Thus, each of the independent variables ($W_{1}$ to $W_{6}$) had its effect quantified by the model. To define the coefficient values, the most common method is the least squares means, that minimises the distance of the fitted model to all input data observations. The stochastic nature of the model, in this case, is represented by the residual error term (ϵ) that adds a Normal distribution perturbation to the model outcome. Another important aspect of the statistical modelling is that the values of the parameters themselves have an uncertainty. In the frequentist approach, this uncertainty is due to the sampling/experimental very nature and it is assessed with confidence intervals. The confidence interval is important to define the significance of the coefficient to the model and, consequently, to decide if the variable should be included in the final model. In the Bayesian approach, this uncertainty comes from the fact that the parameters themselves have their own probability distributions and it is assessed by the credibility intervals [27]. Examples of the statistical modelling in veterinary field are Perri et al. [28], which investigated the risk factors for porcine epidemic diarrhea in Canadian swine herds through a Generalized Linear Model (GLM), and Esteban-Gil et al. [29], which studied the bovine besnoitiosis patterns with multivariable logistic regression models.

There are few applications of modelling techniques to CAE disease studies and the purpose of the present work is to review them, highlighting their scope, strengths and limitations, as well to provide ideas for future modelling approaches for CAE disease.

The present work is organised as follows: the Section 2 present the published articles that use epidemiological models for CAE; Section 3 present articles that used statistical models to investigate the disease risk factors, the consequences of SRLV infection on goats diary production and CAE as a risk factor for other goat diseases. Section 4 present the main conclusions and recommendations for future work.

## 2. Epidemiological Models

The epidemiological modelling studies for CAE are based on the SIR model (Equation (Equation 1)). In this model each compartment is represented by a differential equation that models the variation of the number of individuals in the corresponding compartment with time. The Susceptible and the Infected compartments are self-explanatory and the Removed division is composed by deceased or recovered individuals. Since there is no cure for CAE, it does not make sense to model the Recovered compartment and the models discussed in this work have only the Susceptible and the Infected compartments. Other than that, the original SIR model was defined for a short-term infection, so it does not have births or mortality terms. Because CAE is a long term disease, they become important and are included into the studied models. The main distinction between the epidemiological models for CAE proposed so far in the literature refers on the conceptual model adopted to describe the horizontal transmission paths. One group of models considered only the sexual contact while the other group considered only the direct contact as the horizontal transmission path.

### 2.1. Sexually Transmitted Disease Models

The models proposed in [30,31,32,33] articles are based on SIR models for AIDS and the only horizontal transmission path included in these models is the sexual transmission. These articles are all of the same study group and the authors justify the modelling of the sexual transmission for CAE saying that “there is a possibility that there is an infection by mating”. Nevertheless, this affirmation is based on a single study [15] that founded the pro-lentiviral DNA in male sexual organs of small ruminants and intermittent shedding of proviral SRLV DNA into ejaculated semen. Following the concept of a sexually transmitted disease, in these models the population is divided in Juvenile, Male and Female subpopulations and each subpopulation has the Susceptible and Infected compartments.

Hirata et al., 2013 (I) and Hirata et al., 2014 (II)

The epidemiological model proposed in [30,33] is defined by:(4)$$\left\{\begin{array}{c} J_{S}^{\prime }\;=\;\left(1-e\right)b-\mu_{J}J_{S}-gJ_{S}\\ J_{I}^{\prime }\;=\;eb-\left(\mu_{J}+\delta \right)J_{I}-gJ_{I}\\ M_{S}^{\prime }\;=\;\frac{1}{2}gJ_{S}-\mu_{M}M_{S}-\beta_{M}\frac{F_{I}}{F_{S}+F_{I}}M_{S}\\ M_{I}^{\prime }\;=\;\frac{1}{2}gJ_{I}-\left(\mu_{M}+\delta \right)M_{I}+\beta_{M}\frac{F_{I}}{F_{S}+F_{I}}M_{S}\\ F_{S}^{\prime }\;=\;\frac{1}{2}gJ_{S}-\mu_{F}F_{S}-\beta_{F}\frac{M_{I}}{M_{S}+M_{I}}F_{S}\\ F_{I}^{\prime }\;=\;\frac{1}{2}gJ_{I}-\left(\mu_{F}+\delta \right)F_{I}+\beta_{F}\frac{M_{I}}{M_{S}+M_{I}}F_{S} \end{array}\right.$$
where ${}_{S}$ and ${}_{I}$ are indexes to the Susceptible and Infected compartments, respectively; *J*, *M* and *F* are the juvenile, male and female subpopulations; *e* is the vertical transmission rate; *b* is the total number of births; μ is the natural mortality rate; *g* is the goats growth rate; δ is the disease-related mortality rate; and $\beta_{M}$ and $\beta_{F}$ are the horizontal infection rate for male and female subpopulations. Notice that, by definition, these infection rates refer to the infection rate *stricto sensu* for sexual contacts. However, they also weight the full combination of sexual contacts from all susceptible with all infected goats, expressed by the term with the multiplication of the susceptible by the proportion of infected individuals from the other gender.

In this epidemiological model, goats enter the juvenile compartment being born and they leave it by death or growing up. Importantly, there is no differentiation between the newborns from infected or from susceptible goats. As this is not explicit, to handle with this the vertical infection parameter (*e*) should change according to the size of female susceptible subpopulation. Half of the juvenile will enter the male subpopulation and half will enter the female subpopulation, into the susceptible or infected compartments according to their CAE status when juvenile. The goats leave the male and female subpopulations also by death or they can leave the susceptible and enter the infected compartments of the corresponding subpopulations if they become infected. Another aspect of this model that is worth to mention is that the disease-related mortality for juvenile, male and female subpopulations is the same. However, the clinical symptoms of CAE, and consequently their intensity, are related to the duration of the infection. Thus, it would be expected that the disease-related mortality for adults would be higher than from juvenile.

According to the authors in [30], the improvement of this model should include a mortality parameter specific for goats slaughtered because of their seropositive CAE status and by considering the seasonality of the sexual intercourse in goats (that happen once or twice a year). This last consideration reinforces the perspective of CAE as a sexually transmitted disease, which is not well supported by the literature.

Hirata et al., 2013 (II)

The epidemiological model proposed in [31] (Equation (Equation 5)) follows the previous one (Equation (Equation 4)), with three differences. These differences are in bold in the model definition (Equation (Equation 5)). The first is in the term that models the inflow of juvenile by being born. Now the model distinguishes the CAE status of the mothers. So, in this second model, the vertical infection rate (*e*) accounts truly only for the vertical infection rate. The second difference is that the natural mortality rate for juvenile ($\mu_{J}$) and adults subpopulations (μ) are different. The last difference is the inclusion of a sexual prevention rate (*d*) in the sexual contacts term. This parameter indicates once more the influence of the AIDS humans model in the proposed CAE model, as the authors do not mention any technique for preventing sexually transmission of diseases in goats mating.
(5)$$\left\{\begin{array}{c} J_{S}^{\prime }\;=\;\eta F_{S}+\left(1-e\right)\eta F_{I}-\mu_{J}J_{S}-gJ_{S}\\ J_{I}^{\prime }\;=\;e\eta F_{I}-\left(\mu_{J}+\delta \right)J_{I}-gJ_{I}\\ M_{S}^{\prime }\;=\;\frac{1}{2}gJ_{S}-\mu M_{S}-\left(1-d\right)\beta_{M}\frac{F_{I}}{F_{S}+F_{I}}M_{S}\\ M_{I}^{\prime }\;=\;\frac{1}{2}gJ_{I}-\left(\mu +\delta \right)M_{I}+\left(1-d\right)\beta_{M}\frac{F_{I}}{F_{S}+F_{I}}M_{S}\\ F_{S}^{\prime }\;=\;\frac{1}{2}gJ_{S}-\mu F_{S}-\left(1-d\right)\beta_{F}\frac{M_{I}}{M_{S}+M_{I}}F_{S}\\ F_{I}^{\prime }\;=\;\frac{1}{2}gJ_{I}-\left(\mu +\delta \right)F_{I}+\left(1-d\right)\beta_{F}\frac{M_{I}}{M_{S}+M_{I}}F_{S} \end{array}\right.$$
where η is the birth rate; *d* is the sexual prevention rate; and the other terms follow the parameters of the Equation (Equation 4). The bold symbols are those that were modified or introduced from the previous model (Equation (Equation 4)).

Hirata et al., 2014 (I)

The last epidemiological model in the sexually transmitted diseases class was proposed in [32]. This model (Equation (Equation 6)) is quite similar to the one depicted in Equation (Equation 4), but it considers the seasonality of the sexual contacts of the goats with a cyclical sexual contacts rate parameter ($c_{M}$ and $c_{F}$ for male and female subpopulations, respectively). As a consequence of this seasonality of sexual contacts, the model also considers a seasonality of the goats births.
(6)$$\left\{\begin{array}{c} J_{S}^{\prime }\;=\;\left(1-e\right)B-\mu_{J}J_{S}-gJ_{S}\\ J_{I}^{\prime }\;=\;eB-\left(\mu_{J}+\delta \right)J_{I}-gJ_{I}\\ M_{S}^{\prime }\;=\;\frac{1}{2}gJ_{S}-\mu_{M}M_{S}-\beta_{M}\;c_{M}\frac{F_{I}}{F_{S}+F_{I}}M_{S}\\ M_{I}^{\prime }\;=\;\frac{1}{2}gJ_{I}-\left(\mu_{M}+\delta \right)M_{I}+\beta_{M}\;c_{M}\frac{F_{I}}{F_{S}+F_{I}}M_{S}\\ F_{S}^{\prime }\;=\;\frac{1}{2}gJ_{S}-\mu_{F}F_{S}-\beta_{F}\;c_{F}\frac{M_{I}}{M_{S}+M_{I}}F_{S}\\ F_{I}^{\prime }\;=\;\frac{1}{2}gJ_{I}-\left(\mu_{F}+\delta \right)F_{I}+\beta_{F}\;c_{F}\frac{M_{I}}{M_{S}+M_{I}}F_{S} \end{array}\right.$$
where B is the cyclical total births (Equation (Equation 9)) and the $c_{M}$ and $c_{F}$ (Equation (Equation 7)) are, respectively, the cyclical sexual contact rate for male and female subpopulations. The bold terms indicated the modified terms from model Equation (Equation 4).

The definition of the cyclical sexual contact rate (c), both for male and female subpopulations, is given by:(7)$$\begin{array}{c} c=\left(\frac{1}{p}\right)\left(e^{a_{c}}-1\right)\\ a_{c}={\left[\;\left|sin\left(2\pi T^{-1}\left(t-\theta \right)\right)\right|-sin\left(2\pi T^{-1}\left(t-\theta \right)\right)\;\right]}^{2} \end{array}$$
where *p* is the cycle peak parameter; *T* is the breeding parameter; and Θ is the cycle phase parameter.

And the total births parameter (B) is defined by:(8)$$B=\eta (F_{S}+F_{I})$$
where η is the cyclical birth rate defined by:(9)$$\begin{array}{c} \eta =\left(\frac{1}{p_{\eta }}\right)\left(e^{a_{\eta }}-1\right)\\ a_{\eta }={\left[\;\left|sin\left(2\pi T^{-1}\left(t-\theta +150\right)\right)\right|-sin\left(2\pi T^{-1}\left(t-\theta +150\right)\right)\;\right]}^{2} \end{array}$$

Notice that the cyclical birth rate have the same cyclical pattern of the cyclical sexual contact rate (Equation (Equation 7)), with a phase delay of 150, that is the average gestation days of the goats.

In [30,33], the authors presented the equilibra analysis of the model (Equation (Equation 4)) for the disease-free and the endemic equilibra points. The disease-free equilibrium was proved to be asymptotically stable and the endemic equilibrium was proved to be locally asymptotically stable. However, the consequences of the conditions of these equilibra points were not explored by the authors in terms of the impact of the system parameters for the disease spreading or contention.

These articles [30,31,32,33] also present computational simulations using the proposed models. The main features of these simulations are summarised in the Table 1. In [30,31], the authors used for ODEs numerical simulations both the Euler and the Runge-Kutta iterations methods, while in [32], they used just the Euler method and they do not mentioned which method they used in [33]. In both articles from 2013 they analysed just the endemic scenario, while in the articles from 2014 the authors analysed both the disease-free and the endemic scenarios in their simulations. Besides that, in [30] the authors considered the seasonality of the births in the simulations and in the [32] they considered the seasonality of the births and of the sexual contacts in the simulations. The parameters values used in the simulations were not justified, with exception of the natural mortality and growth rate, which were based in field knowledge.

In all simulations, the authors pointed that good convergence was observed, but the plots presented in the articles do not show it undoubtedly. The simulation results in [30] compare the scenario with and without seasonality of the goat births and the authors highlight how important this difference is based only in the difference itself, without comparing it with any real data. In [31] the only comparison the authors discuss on their simulations is with regard to the use of the Euler first order iterations method or the Runge-Kutta fourth order iteration method. The results were quite similar and the authors justify it saying that the time step of one day used in the simulations was small in respect to the long span of livestock life cycle from birth to slaughter. Hirata T. et al. (2014) [32,33] the simulations compared the disease-free and the endemic scenarios. In the former article, the sexual contacts and the births seasonality were used and the authors affirmed that there is an important reduction of the goats population in the presence of the CAE disease. However, this decrease in the population size depends on the values of the parameters used in the simulations and these values were not compared or justified. In the later article, the authors pointed that the infected compartments increased fast. Again, it depends on the values of the parameters used and it is not discussed it in terms of the real-world values or on how these simulations can help to plan contingency or eradication measures. Actually, in [31] the authors propose a design of a quarantine system of goats applying the proposed epidemiological model for CAE in Tarama Island (Japan). In the presented design, the authors suggest the use of the model on the quarantine software, responsible to predict epidemic infections based on online real data monitoring. But any result of the use of their model in this system is presented.

Finally, the [30,32,33] articles also proposed a definition for the basic reproduction rate ($R_{0}$) for CAE, based on their models. The basic reproduction rate is defined as the expected number of secondary cases produced by a single (typical) infection in a completely susceptible population [34] and it is used to measure the potential of infection of an infectious disease. There are many strategies to calculate it [35], but the basic definition in the epidemiological models [36] is given by:(10)$$R_{0}\;=\;\frac{\alpha }{\lambda }$$
where α is the transmission rate, the inflow into the Infected compartment; and λ is the recovery rate, the outflow of the Infected compartment.

In [30,33], the definition of $R_{0}$ is valid only for the juvenile subpopulation (Equation (Equation 11)):(11)$$R_{J,0}\;=\;\frac{be}{\mu_{j}+g+\delta_{j}}$$
where be is the inflow rate and $\mu_{j}+g+\delta_{j}$ is the outflow rate of the Infected compartment.

In [32], the proposed $R_{0}$ of CAE is defined for the entire population:(12)$$R_{0}\;=\;\frac{\frac{\beta_{M}M_{S}+\beta_{F}F_{S}}{M_{S}+F_{S}}c+e\eta }{\frac{\mu_{J}J+\mu_{M}M+\mu_{F}F}{J+M+F}+\delta }$$

According to the authors, as the CAE infection rate does not depend on the ratio of the sex, the $\frac{F_{I}}{F_{S}+F_{I}}$ and $\frac{M_{I}}{M_{S}+M_{I}}$ terms are not necessary in the $R_{0}$ definition. However, they do not justify why the sexual contact rate *c* used for the $R_{0}$ definition was the same for male and female subpopulations, while in the epidemiological model they were different. In addition, the rates of the denominator are weighted by the total of individuals in the juvenile, male and female subpopulations. Nevertheless, given that the $R_{0}$ is the relation between the inflow and the outflow rates of the infected compartments [37], these weights should have considered only the infected compartments.

Generally speaking, these articles [30,31,32,33] are speculative studies on epidemiological models that can be applied to CAE. They lack discussion on how the models can be interpreted or how the models can improve the understanding of the disease spread and, consequently, its contention or eradication.

### 2.2. Direct Contact Transmitted Disease Models

Two other articles from literature use the differential equations SIR model to model CAE dynamics, [38,39]. Instead of considering CAE as a sexually transmitted disease, they consider the direct contact as the only horizontal infection path. In addition, these models are SRLV genotype-specific and they include in the model an important characteristic of the CAE, the long incubation time. For that, the Infected compartment is divided into infected asymptomatic and infected symptomatic compartments.

Pittavino et al., 2014

In [38] the authors explicitly model CAE infection only for the lentivirus genotype B. The article has two goals: (i) to evaluate the replacement rate (the proportion of the newborns that need to be raised in order to maintain the total goat population constant) in disease-free and in infected herds and (ii) to study under what parameter combinations the disease could disappear or become endemic.

First, the authors define the replacement rate for a disease-free breeding:(13)$$\alpha =\frac{\mu }{r}$$
where μ is the natural mortality rate and *r* is the reproduction rate, given by:(14)r=fpl
where *f* is the fertility rate, the proportion of pregnant goats; *p* is the reproductivity rate, the number of newborns with respect to the number of pregnant goats; and *l* is the live birth rate, the proportion of live newborn goats. Using values from literature, the authors evaluated the reproduction rate for three breeds in a disease-free scenario: the Sardinian Race (≅9.0%), the Roccaverano Race (≅6.5%) and the Saanen Race (≅7.3%).

Then, the epidemiological model is defined by:(15)$$\left\{\begin{array}{c} S^{\prime }\;=\;\alpha rS+\left(1-\gamma \right)\alpha r\left(I_{a}+I_{s}\right)-\mu S-\beta S\left(I_{a}+I_{s}\right)\\ I_{a}^{\prime }\;=\;\beta S\left(I_{a}+I_{s}\right)+\gamma \alpha r\left(I_{a}+I_{s}\right)-\left(\delta +\mu \right)I_{a}\\ I_{s}^{\prime }\;=\;\delta I_{a}-mI_{s} \end{array}\right.$$
where *S*, $I_{a}$ and $I_{s}$ are, respectively the Susceptible, Asymptomatic Infected and the Symptomatic Infected compartments; γ is the rate of newborns infected by the mother; β is the rate of contacts between susceptible and infected goats; δ is the progression rate from asymptomatic to symptomatic infected; and *m* is the disease-related plus natural mortality.

According to this model, the goats enter the Susceptible compartment by being born from susceptible mothers or from infected mothers if isolated from them. And the goats leave this compartment by natural death or by becoming infected, with a probability proportional to the contact rate with infected goats. They enter the Asymptomatic Infected compartment by becoming infected or by being born from infected mothers if not isolated from them, and they leave this compartment by dying from natural causes or by the progression to symptomatic infection status. Finally, they enter the Symptomatic Infected compartment by the clinical symptoms progression of the disease and they leave it by natural or disease-related death.

From this model, the replacement rate for an infected breeding is defined as:(16)$$\alpha_{path}=\frac{\mu }{r}+\frac{m-\mu }{r}i_{s}=\alpha +\frac{m-\mu }{r}i_{s}$$
where $i_{s}$ is the proportion of symptomatic infected goats in a given time. Thus, according to the relation between the disease-related mortality and the reproduction rate for the Symptomatic infected compartment, the article define how much larger the replacement rate should be in the presence of CAE infection.

The equilibra analysis of Equation (Equation 15) considers the disease-free and the endemic equilibrium points and the stability of the equilibra points is proved to be dependent of the $D_{\gamma }$ (Equation (Equation 17)) and $N_{\gamma }$ (Equation (Equation 18)) quantities.
(17)$$D_{\gamma }\;=\;\frac{\delta \gamma (m-\mu )}{\beta (m+\delta )}$$
(18)$$N_{\gamma }\;=\;\frac{m(\mu +\delta )}{\beta (m+\delta )}\;-\;\frac{\mu \gamma }{\beta }$$

The disease-free equilibrium is locally asymptotically stable for $N<N_{\gamma }$, where N is the herd size, considered constant because, in practice, the size of the herd do not change over time. If $N_{\gamma }>N>D_{\gamma }$, the endemic equilibrium is infeasible, but if $N<D_{\gamma }$, it becomes feasible but unstable. While for the condition $N>N_{\gamma }$ it is not only stable, but becomes feasible. As this $N>N_{\gamma }$ condition is the opposite condition for the disease-free equilibrium ($N<N_{\gamma }$), the authors pointed that there is a transcritical biffurcation point that depends on $N_{\gamma }$. This means that for any given value of the breeding population only one of the two equilibra can be reached. Therefore, still according to the authors, in these conditions, as the endemic equilibrium is globally asymptotically stable, the disease remains endemic and all trajectories tend to this equilibrium. Additionally, the authors present some computational simulations to study graphically the $N_{\gamma }$ threshold, varying the values of the β, δ and γ parameters. This study shows that the disease-free equilibrium is stable only for values of *N* so small that the breeding population is not commercially practicable.

The authors conclude mentioning that the equilibra analysis of Equation (Equation 15) shows that even if in principle there could be the possibility of eradication of CAE (the disease-free equilibria is locally asymptotically stable), the endemic equilibrium is prevalent and, therefore, the measures aimed at culling the infected goats and removing newborns from infected mothers are only possibilities to keep the epidemics in check.

Venturino et al., 2019

In [39], the model deals with lentivirus genotypes B and E (endemic in Roccaverano, Italy) assuming that the goats are equally susceptible to both, but that the goats cannot be infected by both simultaneously. Besides that, the genotype E infected goats remain assymptomatic and this genotype is assumed to be transmitted only through colostrum, based on the literature about this type of SRLV transmission. Because of this configuration, the model (Equation 19) has four compartments instead of three: Susceptible, Genotype-B Asymptomatic Infected, Genotype-B Symptomatic Infected and Genotype-E Infected.
(19)$$\left\{\begin{array}{c} S^{\prime }=\left\{\;\left(1-\gamma \right)+\gamma \;\left[1-\left(\theta_{B}\;\frac{I_{a}+I_{s}}{N}+\theta_{E}\;\frac{Y}{N}\right)\right]\;\right\}\alpha rN\;-\;mS\;-\;\beta S\frac{I_{a}+I_{s}}{N}\\ I_{a}^{\prime }=\;\theta_{B}\frac{I_{a}+I_{s}}{N}\;\gamma \alpha rN\;+\;\beta S\frac{I_{a}+I_{s}}{N}\;-\;\left(\delta +m\right)I_{a}\\ I_{s}^{\prime }=\;\delta I_{a}-\mu I_{s}\\ Y=\;\theta_{E}\;\frac{Y}{N}\;\gamma \alpha rN\;-mY \end{array}\right.$$
where *S*, $I_{a}$, $I_{s}$ and *Y* are, respectively the Susceptible, Genotype-B Asymptomatic Infected, Genotype-B Symptomatic Infected and the Genotype-E Infected compartments; γ is the probability of not being isolated by the infected animals; $\Theta_{B}$ and $\Theta_{E}$ are the probability of the transmission through colostrum by genotype B and genotype E, respectively; *N* is the herd size; *m* is the natural mortality; μ is the disease-related mortality; and the other parameters follow the definitions of the previous model (Equation (Equation 15)).

According to this model, goats enter the Susceptible compartment by either being removed from their infected mothers and raised in an isolated breeding, with probability 1−γ, or if even not being immediately removed they are not infected by neither of the two genotypes, with probabilities $\Theta_{B}$ and $\Theta_{E}$ respectively. They leave this compartment by natural mortality or by becoming infected with genotype B by direct the contact with infected ones. They enter the Genotype-B Asymptomatic Infected compartment by being infected by infected mothers or by direct contact with infected ones and they leave this compartment by the progression of the symptoms or if they die by natural causes. They enter the Genotype-B Symptomatic compartment by the progression of the symptoms of the asymptomatic infected goats and they leave it by natural or disease-related mortality. Finally, they enter the Genotype-E Infected compartment by becoming infected by genotype-E seropositive mothers and they leave it by natural mortality.

The authors present the analysis of three equilibra points for the model: the disease-free, the genotype-E-free and the endemic equilibra, the first two analytically and the later numerically. From the conditions analysed, only one of the equilibra points can exist. The main conclusions from these analyses from the authors are:–The disease-free equilibra is limited to strict conditions (as also found by [38]) and do not depend on the value of the probability of not being isolate from the infected mother (γ), except for small values of the probability of contact between asymptomatic and symptomatic goats (β).–When the isolation of newborns from their mothers is efficient (small γ), the population which represents genotype-E infected goats vanishes while the genotype-B infected population grows. However, if the isolation of newborns from their infected mothers is less effective (large γ), the population of genotype-E infected goats grows quickly, while the genotype-B infected subpopulation decreases. This result is dependent of the model assumption that the goats cannot be infected by the two SRLV genotypes at the same time, but it is consistent with what was observed in the Roccaverano farms, where the genotype B is not present even in the absence of disease containment measures by the farmers.–If the probability of not being isolated from infected mothers (γ) exceeds a certain threshold, the endemic equilibrium becomes feasible and asymptotically stable, arising via a transcritical bifurcation.–As the two strains are transmitted vertically but the genotype-B is the only one transmitted horizontally, the higher the probability of not being isolated (γ), the greater the chance of existence of the endemic equilibrium;–The higher the rate of progression of the symptoms (δ), the smaller the range of values of the rate of contacts between symptomatic and asymptomatic goats (β) for which the endemic equilibrium can exist.

Based on these conclusions from the model, the authors suggest that in the presence of both genotypes and when the only control measure is the mother-newborn separation, a complete reversal of the current raising policy should be performed. In this case, the farmer should not isolate the newborns from their mothers but rather let them be raised with all the other animals in the farm and favour instead their high mixing. On the other hand, in case of an only-genotype B-affected farm, serological testing and mother-offspring separation should be still considered the best strategy for CAE control.

The articles [38,39] present a much more consistent mathematical analysis than the previous epidemiological models and relate the models findings with actions that can help in CAE control.

## 3. Regression Models

In this section we present the studies where statistical modelling was applied to investigate the influence of CAE in goat milk and cheese productions, risk factors and in the incidence of other diseases in goats. These studies are presented in groups according to the main scope of the publication.

### 3.1. Diary Production Models

The milk and diary production is the main motive for goats breeding in the world. Therefore, to understand how the incidence of CAE quantitatively affects the milk quality and production is fundamental. The models presented in this section have this goal. They are summarised in Table 2 and explained bellow.

Nord and Adnoy, 1997

In this article [40], the effects of CAE on lactational performance of goats is investigated through statistical modelling. Mean production of milk, protein, fat, and lactose and Somatic Cell Counts (SCC) were compared for seropositive and seronegative goats. The animals included in the study were randomly selected according to the total age distribution in each herd. The sampling was made in two periods and each one was considered a different lactation period. Two models are considered in the article:(20)YMP=age+herd+CAE+CAE×age+ϵ
where YMP is the year milk production; age is the goat’s age in years; CAE is the goat’s CAE status; and ϵ the residual error.
(21)Y=D+D×D+log(D)+log(D)×log(D)+age+CAE+CAE×age+h+cdr+gch+ϵ
where *Y* can be the daily milk production, fat weight percentage, protein weight percentage, lactose weight percentage the the log of SCC; *D* is the day of lactation over 365 days; *h* is the herd; cdr is the control date of the herd; gch is the goat within CAE test result and herd.

Importantly, the model of this study considers three CAE status: negative, positive and indeterminate. In other words, an unknown CAE status is modelled as a level of the CAE seroprevalence factor. However, this choice is questionable, as the meaning of an unknown status is completely different from the meaning of a positive or a negative status. If considering an unknown CAE status, the other possible status with the same meaning would be the known status, independently if it would be positive or negative. Despite that, in a biological modelling perspective, it is hardly justifiable to say that the production of milk or the quality of the milk would depend on the human knowledge of the CAE status of the goats.

The effect of the CAE status on annual milk production was not statistically significant, but the interaction of CAE status and age was significant. At 5 years of age, seropositive goats produced more milk than seronegative goats or goats that had an indeterminate response to the CAE test (*p*-value = 0.03 and 0.001, respectively), and seronegative goats produced more milk than goats with the indeterminate CAE status (*p*-value = 0.02). According to the authors, the possibly larger mean of the milk production for seropositive goats might be a consequence of an increased immune response by goats that are infected with CAE or it can indicate that high producing goats may also be more susceptible to infection from CAE. These affirmations, however, are not supported by data or by studies from the literature and, actually, do not correspond to any other study found in literature. Indeed, the affirmation that high producing goats may also be more susceptible to infection from CAE seems to be a spurious correlation case.

Another result reported in this article is that the difference between percentages of milk fat of seronegative and seropositive goats was not significant (*p*-value = 0.19), while the percentage of fat in the milk of goats with indeterminate antibody results was significantly lower than the percentage of fat in the milk of the seronegative goats (*p*-value = 0.007). Still according to the authors, the lack of differences in milk production and percentages of fat, protein, and lactose in milk between seropositive and seronegative herdmates indicates that CAE infection in otherwise healthy goats do not influence lactation performance to any great extent.

An overall statistically significant interaction between CAE infection and age was found for SCC (*p*-value = 0.03). The SCC for seropositive goats increased from first to second lactation (age, therefore) (*p*-value = 0.06). But for seronegative goats the SCC decreased from first to second lactation (*p*-value = 0.06). During the first lactation, the difference was not significant (*p*-value > 0.05) between goats that were positive or negative for CAE status. However, during the second lactation, seropositive goats had slightly higher SCC than did seronegative goats (*p*-value = 0.05). The SCC was increased for seropositive 2 year old goats, indicating an effect of CAE on milk quality and on pathological changes in the udders, even of young goats.

Even with a high number of goats in the study, the interpretability of the models were not straightforward, and the use of the indeterminate CAE status negatively influenced the quality of the presented results.

Martinez-Navalón et al., 2013

In this article [41], the authors used the statistical modelling to investigate the production and the quality of the milk in CAE infected goats. To build the final model, they use a backward-elimination strategy with the Akaike’s information criterion (AIC). For the daily milk yield (in litres), the final model is in Equation (Equation 22):(22)DMY=PS+NOLS+LM+LD+CAE×PN+H+G+ϵ
where DMY is the daily milk yield, PS the fixed effect parity season, NOLS the fixed effect number of offspring in the last kidding, LM the fixed effect lactation month, CAE×PN the fixed effect interaction between CAE status and parity number, *H* the random effect of the herd, *G* the random effect of the goat, and ϵ the random residual.

The significant effects of this model were:–parity season: winter was associated with the highest milk production, followed by autumn, spring and summer, in this order;–number of offspring in the last kidding: 2, 3 or more kids in the last kidding was associated with the higher milk production, if compared with 1 kid; otherwise, the milk production was not significantly different if the goat gave born to just 1 kid or if it aborted the last gestation;–lactation month: except for the first lactation month, in all other lactation months the milk production was smaller when compared to the second lactation month;–lactation duration: lactation duration between 63 and 211 days produced less milk;–interaction between CAE status and parity number: the daily milk yield for seronegative goats in the third lactation was 9.6% bigger than for seropositive goats in the same lactation, and it was 5.4% and 16.7% bigger for the sixth and greater lactations, respectively.–the herd and the goat random effects accounted for 49.7% and 7.3% of the variation of the response variable.

The authors also reported that regression models confirmed the strong significant association between CAE seropositivity and reduced percentages of milk fat and dry extract, but not for percentage of milk protein.

Nowicka et al., 2015

In this article [42], the authors applied a statistical model to study the effect of CAE infection on cheese production (in terms of the amount of fresh cheese obtained from 1 kg of milk) in goats. The study was carried out in Poland in just one herd, during three years but only with 63 dairy goats. The goats were tested to SRLV at the age of 4–6 months and then twice a year. Goats with seroconverted status either during the observation period or during a year preceding the onset of the study were excluded from the analysis.

The final model included goat and age/parity as independent variables (kept in the model independent of the *p*-value of their coefficients to handle with the dataset unbalance and with the age difference between SRLV groups, respectively); protein content, fat content, 3rd stage of lactation as confounding variables (*p*-value of the coefficients all <0.0001); and the SRLV infection (*p*-value 0.013).

The final model was defined by:(23)Y=−0.58+0.02goat+0.27age+1.34prot+0.89fat−8.973rdlac−4.6SRLV+ϵ
where *Y* is the milk production; goat is the goat; age is the goat age; prot is the protein content of the milk; fat is the fat content of the milk; 3rd lac is the third lactation fixed effect; SRLV is the SRLV status of the goat; and ϵ is the residual error.

According to the authors, their results indicate that SRLV infection may reduce the amount of cheese obtained from a goat. Moreover, this seems to occur not only through reduction of milk component concentration but also via other mechanisms, deteriorating the cheese-forming properties of milk. Although statistically significant, the extent of cheese yield reduction was very small. Keeping constants the stage of lactation, the protein and fat milk contents, the mean cheese yield was lowered on average by 4.6 g cheese per 1 kg of milk in a seropositive goat compared with a seronegative one.

The statistical modelling of this article is very good. However, it could have been interesting to model not only the cheese production, but the milk yield and contents too. It would help to understand the confounding variables effects, since the independent variables could be associated (SRLV status and protein or fat milk contents can be correlated). Additionally, the authors mention that the modelling may have been compromised by the small number of goats in the dataset and by the difference in goats age between positive and negative SRLV infection groups.

### 3.2. CAE Risk Factors Models

Kaba et al., 2012

The only article that used statistical modelling to investigate CAE risk factors was Kaba et al. [43]. Actually, the main data analysis of the article is made with the Student’s *t*-test to analyse the CAE infection on the quantitative and qualitative characteristics of milk production in dairy goats. However, the authors also build a statistical model to analyse the risk factors for a CAE infection. Even with a long-term cohort study, data in [43] refer to a single herd in which, during the 12 years, they had a total 177 goats. This is handled in the model with the animal additive random effect, but it is important to notice that the 12 years long do not refer to 177 goats, since the herd size was of approximately only 50 goats. Their model is defined as follows:(24)$$Y_{ijklmno}=a_{i}+p_{i}+ys_{j}+td_{k}+SS_{l}+LS_{m}+P_{n}+\beta \left(x_{ijklmno}-x\right)+{\left(\Sigma bpLPd^{p}\right)}_{ijklmno}+\epsilon$$
where $Y_{ijklmno}$ is the serological status of the goats; $a_{i}$ is the animal; $p_{i}$ is the permanent environment effect; $ys_{i}$ is the year-season of kidding random effect; $td_{k}$ is the date of the test; $SS_{l}$ is the CAE status (positive or negative; they excluded from the study the goats with inconclusive CAE test result); $LS_{m}$ is the litter size (1, 2 or 3); $P_{n}$ is the parity (1, 2 or 3), $\beta (x_{ijklmno}-x)$ is a fixed effect regression on milk yield for fat, protein and lactose contents and SCC; ${\left(\Sigma bpLPd^{p}\right)}_{ijklmno}$ is a fixed effect regression of Legendre polynomials of standardized days in milk (1, 2, 3 or 4); and ϵ is the residual error.

According to the results presented in the article, the serological status of a goat is linked to its parity: the higher the parity, the greater the probability of CAE infection; the probability of infection in 1 year old goats was lower by 84% and in 2 years old goats by 42% when compared with goats aged 3 or more years (*p* ≤ 0.01).

### 3.3. CAE as Risk Factor for Other Diseases Models

This section presents studies that modelled the effect of CAE as a risk for other diseases. They are summarised in Table 3.

Sanchez et al., 2001

In this article [44], the authors investigate the relationship among CAE, intramammary infection (IMI) and SCC in goat milk. The statistical model proposed in the article has the SCC as dependent variable and it is defined as:(25)$$SCC_{ijk}=a+herd_{i}+IMI_{j}+CAE_{k}+{\left(IMI\times CAE\right)}_{jk}+\epsilon .$$
where $SCC_{ijk}$ is the log of SCC; *a* is the intercept; $herd_{i}$ is the herd; $imi_{j}$ is the intramammary infection status of the half of the udder *j*; $CAE_{k}$ is the CAE status of the goat; and ϵ is the residual error.

The CAE infection effect was not significant for SCC, but the interaction between CAE and IMI was significant: Negative IMI and positive CAE increased SCC in respect to Negative IMI but Negative CAE; IMI as a single infection and Positive CAE increased SCC in respect to IMI as a single infection and Negative CAE status; IMI subclinical and positive CAE and IMI subclinical and CAE negative do not had significant effect in SCC; finally, persistent IMI and positive CAE and persistent IMI and negative CAE also do not had significant effect in SCC.

According to the other results presented in this article, the CAE had no apparent effect on IMI and the significant effect of the interaction of CAE and IMI in the SCC model should be considered for diagnosing of subclinical IMI.

Luengo et al., 2004

In this article [45], the authors model the factors influencing the SCC in goats milk for two scenarios: whole-lactation and monthly-lactation average half-udder. Among other factors, they analyse the CAE status and the interaction between CAE status and IMI. However, as can be seen in both models (Equation (Equation 26) for the whole-lactation and Equation (Equation 27) for the monthly-lactation), none of the CAE terms was found significant to the model.
(26)$$Y_{ijlnoq}=\mu +F_{i}+U{\left(F\right)}_{ij}+I_{k}+T_{n}+D_{o}+PL_{q}+\epsilon$$
where $Y_{ijlnoq}$ is the whole-lactation average half-udder log SCC, μ a constant, $F_{i}$ the fixed effect of the flock, $U{\left(F\right)}_{ij}$ the random effect of half udder nested into the flock, $I_{k}$ the IMI status, $T_{n}$ the fixed effect of number of kids born, $D_{o}$ the fixed effect of length of lactation with three levels, $PL_{q}$ the parity and IMI status interaction, and ϵ the random residual.
(27)$$Y_{ijknop}=\mu +F_{i}+U{\left(F\right)}_{ij}+I_{k}+T_{n}+D_{o}+M_{p}+\epsilon$$
where $Y_{ijknop}$ is the montlhy-lactation average half-udder log SCC, μ a constant, $F_{i}$ the fixed effect of the flock, $U{\left(F\right)}_{ij}$ the random effect of half udder nested into the flock, $I_{k}$ the IMI status, $T_{n}$ the fixed effect of number of kids born, $D_{o}$ the fixed effect of length of lactation with three levels, $M_{p}$ the fixed effect of the stage of the lactation, and ϵ the random residual.

Leitner et al., 2010

In this article [46], the authors used the statistical modelling to investigate the effect of CAE on flock production parameters and on udder health. The study was carried out for three years with a total of 248 goats: 118 goats were recorded over at least their first three lactations, 85 goats were recorded over their first and second lactations and 45 goats were recorded only over their first lactation. The CAE serological tests were performed twice over the study period and three CAE serological status were considered: positive for the goats that were positive in both tests, negative for the goats that were negative in both tests, and converted for the goats that were negative in the first test but positive in the second test.

Among the studied herd parameters, the authors define the following model to investigate the effect of CAE status on the number of offspring:(28)$$Y_{ij}=\mu +\alpha_{i}+\epsilon_{1}$$
where $Y_{ij}$ is the number of offsprings; μ is the intercept; $\alpha_{i}$ is the CAE status; and $\epsilon_{1}$ is the residual error. The CAE status was not significant for the number of offspring.

Besides that, the effect of CAE infection on milk quality (based on the California mastitis test (CMY) and SCC) was analysed separately for each lactation by applying the following model:(29)$$Y_{ijklm}=\mu +\alpha_{i}+\epsilon_{1}+\beta_{k}+\alpha \beta_{ki}+\gamma_{l}+\epsilon_{2}$$
where β is the bacterial infection; γ gamma is the difference lactation stage’; and $\epsilon_{2}$ is the global residual. The CAE status was not a significant effect in the incidence of bacterial udder infection. During all lactations, the SCC and CMT were significantly affected by bacterial udder infection and by lactation number, but not by CAE status.

Koop et al., 2013

In this article [47], the authors used a Bayesian logistic regression to evaluate risk factors for the latent IMI status in dairy goats. They used two datasets: one with data from The Netherlands with no CAE infected goats and one with data from California in which about 51% of the goats tested positive or suspect for CAE.

In both datasets, higher parity and lower milk yield were significantly associated with greater odds of infection. The final model for the Californian data included parity, milk yield and sampling occasion, which are uncontrollable risk factors. CAE infection was not included in the final model as a risk factor.

The final model for the full Dutch dataset without the teat, teat-end and udder conformation data was:(30)$$logit\left(IMI_{jlk}\right)=\mu +PAR_{jk}+MY_{jkl}+\gamma_{jkl}+\delta_{k}$$
where PAR is the parity, MY the milk yield, Γ the random effect of the jth goat in the kth herd at the lth sampling and $\delta_{k}$ the random effect of the kth herd.

The final model for the full Dutch dataset including the teat, teat-end and udder conformation data was:(31)$$logit\left(IMI_{jk}\right)=\mu +PAR_{jk}+MY_{jk}+UD_{jk}+\gamma_{jk}+\delta_{k}$$
where $UD_{jk}$ is udder depth and the other variables are the same as defined in the model Equation (Equation 30).

The final model for the Californian dataset was:(32)$$logit\left(IMI_{ijlk}\right)=\mu +SAMP_{ijkl}+PAR_{jk}+MY_{jkl}+\gamma_{jkl}+\delta_{k}$$
where $SAMP_{ijkl}$ is the sampling occasion and the other variables are the same as defined in the model Equation (Equation 30).

## 4. Conclusions

The small ruminant lentivirus epidemiological framework represents a very hard and complex problem in animal management, including a wide range of biosecurity and diagnostic issues. Moreover, in some countries, like in Italy, goats herds are often smaller than the bovine counterparts and they are bred in family farms. These two aspects can strongly influence the animal management, increasing the CAE transmission risks and reducing the biosecurity levels within the farms.

For these reasons, the knowledge about CAE (or SRLV in general) prevalence and distribution is a fragmented patchwork. Precise data about, for example, exposure and transmission probabilities are still lacking, as well as extensive surveillance investigations about the circulating viral genotypes and subtypes. Even if the small ruminant production business is smaller than other economic sectors (i.e., bovine farming), the evaluation of the factors influencing the CAE transmission can strongly improve animal health, wellness and production.

There are few modelling applications for CAE in literature and, on the other hand, there is still a plenty of opportunities for modelling studies on CAE. For example:–Epidemiological models considering the epidemic dynamics not only inside each herd, but also between herds. As goats herds are often bred in family farms, they are less likely to be isolated and the contact of goats from different farms in the pasture can be significant for the CAE spreading;–Epidemiological models considering the housing and pasture scenarios explicitly, since it plays an important role in the horizontal transmission;–The models that investigated the effect of CAE infection on milk and cheese production, have not considered explicitly the progression of the symptoms over the infection time. However, as the progression of the symptoms of CAE is slow, the infection time can be a relevant parameter. Many of them found a positive correlation between the parity number or age and the milk production. These parameters can be significant due a confusion factor, since they are correlated with the infection time;–Milk production models can include the udder condition to investigate if the decrease of milk production in soropositive SRLV goats is due the udder condition or the general health condition of the one;–Other modelling strategies, like stochastic SIR models or Machine Learning (ML) techniques can be explored to investigate the CAE epidemiological dynamic, its risk factors and effects on milk production. Different from the mathematical and statistical models, that are theory-driven, ML models are data driven. This means that the model terms are not defined by a pre-understanding of the system, but they are defined by the algorithm itself. Therefore, this class of models can capture ignored or unknown parts of the system. In particular, since ML models are able to include in the analysis parts of the system that were not considered in the theory-driven models, the combination of these two classes of models can help to clarify the CAE risk factors, epidemics and impacts on milk production.

In conclusion, CAE epidemiology is still a poorly explored world and the actions aimed to better understand transmission dynamics could really improve the animal health, the management system and the production quality. 

## Figures and Tables

**Table 1 animals-11-01457-t001:** Iteration method, equilibra points and seasonalities studied through numerical simulations in [30,31,32,33].

Article	Method	Equilibra	Seasonality
[30]	Euler and Runge-Kutta	Endemic	Births
[31]	Euler and Runge-Kutta	Endemic	
[32]	Euler	Disease-free and endemic	Breeding and births
[33]	Not mentioned	Disease-free and endemic	

**Table 2 animals-11-01457-t002:** Articles that modelled the effects of CAE into dairy products quality and production.

Reference	Data Structure	Statistical Model	Model Scope
Nord and Adnoy, 1997 [40]	Two periods of sampling: 1025 goats sampled from August 1993 to January 1994 and 774 goats from other herds were sampled from August 1994 to Janurary 1995.	Generalized linear mixed model	One model for annual milk production, fat and protein percentages as response variables. Other model for daily milk production, fat, protein and lactose percentage.
Martinez-Navalón et al. 2013 [41]	3913 goats in Valencia that were born from September 2005 and January 2008.	Generalized linear mixed model	To investigated milk production losses associated with serostatus of CAE infection over one lactation.
Nowicka et al. 2015 [42]	247 goats for three years.	Four-level hierarchical linear model	To investigate the influence of small ruminant lentivirus infection on cheese yield in goats.

**Table 3 animals-11-01457-t003:** Articles that modelled the effects of CAE in the incidence of other diseases.

Reference	Data Structure	Statistical Model	Model Scope
Sanchez et al., 2001 [44]	121 goats from 4 herds by 7 months.	Generalized linear mixed model	CAE (among others) effect on SCC.
Luengo et al., 2004 [45]	1304 goat udder halves were sampled monthly during an entire lactation	Generalized linear mixed model	CAE (among others) effect on SCC.
Leitner et al., 2010 [46]	A total of 248 goats of the same herd, being 118 goats for three lactations, 85 for two lactations and 45 for just one lactation	Generalized linear mixed model	The present study was designed to assess the effect of CAE seropositivity on flock production parameters and in particular on udder health. We also looked at the feeding of pasteurised colostrum as a single measure aimed at reducing the spread of CAE infection within goat flocks.
Koop et al., 2013 [47]	530 goats of 5 herds	Bayesian logit model	CAE (among others) as risk factor for intramammary infection modelling. CAE was not selected in the final model.

## Data Availability

Not applicable.

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
