# Peer review of "Caprine Arthritis Encephalitis Virus Disease Modelling Review"

_animals, 2021, doi:10.3390/ani11051457_

Round 1

Reviewer 1 Report

This is a very interesting manuscript which may encourage the scientific society to focus on modelling of lifelong incurable diseases of animals, such as CAE or maedi-visna disease.

Major comments:

CAEV is not a disease but a pathogen (e.g. line 3)! The name of disease is: caprine arthritis-encephalitis (CAE). Please correct here and elsewhere along the text. The abbreviation should be explained when first used.

It’s been accepted for more than decade that the pathogen responsible for CAE is the same as the virus of maedi-visna disease (MVV) and there are named together – small ruminant lentivirus (SRLV), which exists in five (or according to the latest genetic analyses four) genetic groups (genotypes): A, B, C, D (recently joint with A), and E. Please use SRLV instead of CAEV.

Please take note of the fact that I’m not able to go through all equations to ensure that they are correct so please double check them.

Minor comments

Line 64: “ODE” should be explained when first used. Generally, please remember about expanding abbreviations when they are used for the first time.

Line 66: not most simple but the simplest

The references lack names of journals – it’s probably a technical problem but certainly needs to be fixed.

Author Response

Major comments

- CAEV is not a disease but a pathogen (e.g. line 3)! The name of disease is: caprine arthritis-encephalitis (CAE). Please correct here and elsewhere along the text. The abbreviation should be explained when first used.

All occurrences were corrected.

- It’s been accepted for more than decade that the pathogen responsible for CAE is the same as the virus of maedi-visna disease (MVV) and there are named together – small ruminant lentivirus (SRLV), which exists in five (or according to the latest genetic analyses four) genetic groups (genotypes): A, B, C, D (recently joint with A), and E. Please use SRLV instead of CAEV.

The occurrences were corrected.

- Please take note of the fact that I’m not able to go through all equations to ensure that they are correct so please double check them.

The equations were double-checked.

Minor comments

- Line 64: “ODE” should be explained when first used. Generally, please remember about expanding abbreviations when they are used for the first time.

The first occurrence of “ODE" is at line 45 and it is explained.

- Line 66: not most simple but the simplest

The occurrence was corrected.

- The references lack names of journals – it’s probably a technical problem but certainly needs to be fixed.

The references were corrected.

Reviewer 2 Report

Caprine Arthritis Encephalitis Virus disease modelling review

This paper covers a review of statistical and epidemiological (with and without the sexually transmission component) modelling of CAEV and suggestions for future modelling of this incurable disease. Regarding the statistical modelling studies, the reviewed articles varied on modelling assumptions and goals such as dairy production, CAEV risk factors and the hypothesis of CAEV being a risk factor for other diseases.

This review is very worth while as it compares models that are very difficult to compare. It’s main success is lining these models up, explaining them before discussion.

General remarks: The paper is well-written and sections well defined. But, sentences tend to be very long, which does impair readability in general. “Numbers” (references) don’t talk. It seems as if though the paper was written with refences in name and later changes in the requested format of numbers. Please check and adept.

In general the epidemiological models are (well) explained, but minimally discussed, whereas the statistical models are highly discussed.

  1. Introduction

Line 22-23          “The main clinical symptoms of CAEV are encephalomyelitis in juvenile and arthritis in adults.” Encephalomyelitis in juvenile is rare, so I would not call this a main clinical symptom. Furthermore the most prominent symptom is mastitis! And this is not mentioned here. This is a real miss since the statistical models discussed are on “udder output”.

Line 26 “by direct contact (horizontal transmission) in pasture or by animals trading”. In many countries dairy goats are (more or less permanently) housed. This is missing in the equation.

Lines 30-31        “the use of heat-treated 31 colostrum” What is the evidence that heat treated colostrum prevents CAEV transmission? (There are publications accounting for it but more against this claim.)

Line 56                “[20] proposed” Although probably correct, this does not read well. A number (20) that proposes???

Lien 58               “Recently, [22] explored a” See comment above.

Line 59                “. [23] and [24] applied “ Same

General remarks on the introduction section. I like the way the nonspecific CAEV section (lines 39-85) is written. It take the reader by the hand in the land of epidemiology and statistical modelling with veterinary examples.

  1. Epidemiological models

Line 103             “modelsx” Remove x

Lines 217-218    “CAEV infection rate does not depend on the ratio of the sex, the FI FS+FI and MI MS+MI 218 terms are not necessary in the R0 definition”. This is very unlikely in the real world since in goat husbandry generally not all males are kept in with the females after the juvenile stage. Therefore I agree with the authors on their statement in line 223. (No action required here.)

  • Direct contact transmitted disease models

Line 267             “ N is the breeding 268 size” Not sure what is meant by breeding size? (also in line 296)

Line 303             Remove “the”

Line 305             “dye” should be die, as in dead not coloured

Line 321-322     “This result is dependent of the model assumption that the goats cannot 322 be infected by the two SRLV genotypes at the same time”. What is the take of the authors of the review on this assumption?

  1. Regression models

Suggestion: position of table 2 before line 351.

Lines 363-364    “However, this choice is questionable, as the meaning of an unknown status 364 is completely different from the meaning of a positive or a negative status.” I absolutely agree with the authors.

Martinez-Navalón et al. 2013: why is the model not included?

5 3.2. CAEV risk factors models

-

3.3. CAEV as risk factor for other diseases models

Luengo et al. 2004 Why not show the model (even if outcomes negative)?

Koop et al. 2013 Why not show the model

  1. Conclusions

Lines 509-510    “Moreover, 510 goats herds are often smaller than the bovine counterpart and they are bred in family farms” This is not that case in many parts of the world!

It would be interesting if the authors would provide an example “ideal” model based on their review. Or at least which parameters should be included. Suggestions are preferably writing to encourage and stimulate research groups.

Author Response

General remarks

- The paper is well-written and sections well defined. But, sentences tend to be very long, which does impair readability in general. “Numbers” (references) don’t talk. It seems as if though the paper was written with references in name and later changes in the requested format of numbers. Please check and adapt.

We shortened the sentences that we could break up and we changed the citation of the references to use the name of the authors whenever it was possible.

- In general the epidemiological models are (well) explained, but minimally discussed, whereas the statistical models are highly discussed.

The Sexual contacts disease models are very discussed (lines 110, 199, 202, 220, 226) and we included a paragraph with final remarks on Direct contact disease models, since we have little to add to both of these articles.

Introduction

- Line 22-23          “The main clinical symptoms of CAEV are encephalomyelitis in juvenile and arthritis in adults.” Encephalomyelitis in juvenile is rare, so I would not call this a main clinical symptom. Furthermore the most prominent symptom is mastitis! And this is not mentioned here. This is a real miss since the statistical models discussed are on “udder output”.

The occurrence was corrected.

- Line 26 “by direct contact (horizontal transmission) in pasture or by animals trading”. In many countries dairy goats are (more or less permanently) housed. This is missing in the equation.

The models so far published in literature do not consider (at least explicitly) this situation. It was included in the further work.

- Lines 30-31        “the use of heat-treated 31 colostrum” What is the evidence that heat treated colostrum prevents CAEV transmission? (There are publications accounting for it but more against this claim.)

We didn’t find any bibliography reference against this practice. But, instead, many publications cite the use of heat treated colostrum as a prevention path for CAE, e.g., this 2021 SRLV review article:

Wolf C. Update on Small Ruminant Lentiviruses. Vet Clin North Am Food Anim Pract. 2021 Mar;37(1):199-208. doi: 10.1016/j.cvfa.2020.12.003.

Please, can you give us a reference showing that the heat treatment of colostrum is not recommended for CAE prevention?

- Line 56                “[20] proposed” Although probably correct, this does not read well. A number (20) that proposes???

The occurrence was corrected.

- Line 58               “Recently, [22] explored a” See comment above.

The occurrence was corrected.

- Line 59                “. [23] and [24] applied “ Same

The occurrence was corrected.

General remarks on the introduction section. I like the way the nonspecific CAEV section (lines 39-85) is written. It take the reader by the hand in the land of epidemiology and statistical modelling with veterinary examples.

Epidemiological models

- Line 103             “modelsx” Remove x

The occurrence was corrected.

- Lines 217-218    “CAEV infection rate does not depend on the ratio of the sex, the FI FS+FI and MI MS+MI 218 terms are not necessary in the R0 definition”. This is very unlikely in the real world since in goat husbandry generally not all males are kept in with the females after the juvenile stage. Therefore I agree with the authors on their statement in line 223. (No action required here.)

Direct contact transmitted disease models

- Line 267             “ N is the breeding 268 size” Not sure what is meant by breeding size? (also in line 296)

We changed it to “herd size”.

- Line 303             Remove “the”

The occurrence was corrected.

- Line 305             “dye” should be die, as in dead not coloured

The occurrence was corrected.

- Line 321-322     “This result is dependent of the model assumption that the goats cannot 322 be infected by the two SRLV genotypes at the same time.”. What is the take of the authors of the review on this assumption?

The model equilibra conditions reflected this assumption, that determine how the presence of the SRLV E-genotype affected the presence of the SRLV B-genotype. Therefore, we agree with the authors of the reviewed study.

Regression models

- Suggestion: position of table 2 before line 351.

It was not possible, so we positioned it after the Nord and Adnoy, 1997 section.

- Lines 363-364    “However, this choice is questionable, as the meaning of an unknown status 364 is completely different from the meaning of a positive or a negative status.” I absolutely agree with the authors.

- Martinez-Navalón et al. 2013: why is the model not included?

The model was included.

5 3.2. CAEV risk factors models

3.3. CAEV as risk factor for other diseases models

- Luengo et al. 2004 Why not show the model (even if outcomes negative)?

The model was included.

- Koop et al. 2013 Why not show the model

The models were included.

Conclusions

- Lines 509-510    “Moreover, 510 goats herds are often smaller than the bovine counterpart and they are bred in family farms” This is not that case in many parts of the world!

We outlined that this is true only for some parts of the world.

- It would be interesting if the authors would provide an example “ideal” model based on their review. Or at least which parameters should be included. Suggestions are preferably writing to encourage and stimulate research groups.

The authors included one suggestion for modification in epidemiological models (housing) and one new parameter (udder condition) to be considered in statistical models, in addition to the other previously suggested (between-herd dynamic in epidemiological models and symptoms progression in statistical models).